# Chronic Thromboembolic Pulmonary Hypertension: An Update

**DOI:** 10.3390/diagnostics12020235

**Published:** 2022-01-19

**Authors:** Barbara Ruaro, Elisa Baratella, Gaetano Caforio, Paola Confalonieri, Barbara Wade, Cristina Marrocchio, Pietro Geri, Riccardo Pozzan, Alessia Giovanna Andrisano, Maria Assunta Cova, Maurizio Cortale, Marco Confalonieri, Francesco Salton

**Affiliations:** 1Department of Pulmonology, Cattinara Hospital, University of Trieste, 34127 Trieste, Italy; gaetano.caforio@asugi.sanita.fvg.it (G.C.); paola.confalonieri.24@gmail.com (P.C.); pietrogeri@gmail.com (P.G.); riccardo.pozzan@asugi.sanita.fvg.it (R.P.); alessia.g.andrisano@gmail.com (A.G.A.); marco.confalonieri@aots.sanita.fvg.it (M.C.); francesco.salton@gmail.com (F.S.); 2Department of Radiology, Cattinara Hospital, University of Trieste, 34127 Trieste, Italy; elisa.baratella@gmail.com (E.B.); cristinamarrocchio@gmail.com (C.M.); m.cova@fmc.units.it (M.A.C.); 3AOU City of Health and Science of Turin, Department of Science of Public Health and Pediatrics, University of Torino, 10126 Torino, Italy; barbarawade@hotmail.com; 4Department of Medical, Surgical, & Health Sciences, Cattinara Hospital, University of Trieste, 34127 Trieste, Italy; maurizio.cortale@asugi.sanita.fvg.it

**Keywords:** chronic thromboembolic pulmonary hypertension (CTEPH), pulmonary artery pressure (PAP), idiopathic pulmonary arterial hypertension (IPAH), chronic thromboembolism, pulmonary thromboendarterectomy, pulmonary vasodilator therapy

## Abstract

Chronic thromboembolic pulmonary hypertension (CTEPH) is a rare disease observed in a small proportion of patients after acute pulmonary embolism (PE). CTEPH has a high morbidity and mortality rate, related to the PH severity, and a poor prognosis, which mirrors the right ventricular dysfunction involvement. Pulmonary endarterectomy (PEA) reduces pulmonary vascular resistance, making it the treatment of choice and should be offered to operable CTEPH patients, as significant symptomatic and prognostic improvement has been observed. Moreover, these patients may also benefit from the advances made in surgical techniques and pulmonary hypertension-specific medication. However, not all patients are eligible for PEA surgery, as some have either distal pulmonary vascular obstruction and/or significant comorbidities. Therefore, surgical candidates should be carefully selected by an interprofessional team in expert centers. This review aims at making an overview of the risk factors and latest developments in diagnostic tools and treatment options for CTEPH.

## 1. Introduction

Pulmonary hypertension secondary to chronic thromboembolism (CTEPH), a potentially life-threatening condition, is within group 4 of the WHO clinical classification of pulmonary hypertension (PH), based on specific etiological, clinical, diagnostic, and treatment strategies, and takes the form of a distinct pulmonary vascular disease [1,2,3,4,5,6,7]. CTEPH has been frequently reported as causing precapillary PH, characterized by the presence of thromboembolic material with a fibrotic appearance closely adhered to the media of the subsegmental, segmental, and main branches of the pulmonary arteries [1,4,5,6,7,8,9]. Mechanical obstruction of the elastic pulmonary arteries is associated with an increase in pulmonary vascular resistance (PVR), which leads to a pressure increase and ‘shear stress’ which, in turn, leads to arterial disease of the peripheral pulmonary vessels, similar to what can be observed in other forms of precapillary PH [3]. An analysis of the data from a large prospective international registry of CTEPH patients evidenced a history of acute pulmonary embolism (PE) and a history of vein thrombosis in 74.8% and 56.1% of them, respectively [10]. These data highlight the natural history of CTEPH, which appears as a potential long-term complication of one or more thromboembolic events [11,12,13,14]. However, why only a minority of patients (cumulative incidence 0.1–9.1%) may develop PH after an initial episode of PE remains to be clarified [13,14,15,16]. Some epidemiological studies have identified some predisposing factors offering some insight as to the etiology of CTEPH [17,18,19]. Based on available data, it can be stated that CTEPH originates from a remodeling process of the vessel wall after embolic obstruction and is initiated and enhanced by the combination of impaired angiogenesis, reduced fibrinolysis, and endothelial dysfunction. This process may be influenced by predisposing factors, such as infections, autoimmune diseases, chronic inflammatory diseases, hypothyroidism in hormone replacement therapies, oncological diseases, and plasma factors associated with thrombophilic diathesis, and it leads to the progressive development of PH and, subsequently, right heart failure [20,21,22,23,24,25,26].

Diagnosing CTEPH remains a difficult challenge, with an average time of 14 months between the onset of symptoms and diagnosis in expert centers. It involves the confirmation of pre-capillary PH by examination of right cardiac catheterization in patients with perfusion defects of the pulmonary circulation due to a thromboembolic etiology persisting after a period of anticoagulant therapy of at least three months [21,22].

CTEPH differs from other PH groups also as to the therapeutic options currently available [27,28,29]. Pulmonary endarterectomy (PEA) remains the current potentially curative treatment of choice, especially when the organized thrombi involve the main, lobar, or segmental arteries. Although this is a complex surgical procedure involving the removal of obstructive thromboembolic material from the pulmonary vessels, a high percentage of patients have enhanced quality and life expectancy, and it may even be considered a definitive cure for many of them [27,28,29]. Therefore, distinguishing operable from inoperable patients is of crucial importance. Interestingly, a recent study reported that CTEPH female patients treated by PEA had better long-term survival than males [17].

Unfortunately, a certain number of CTEPH (30% to 45%) patients are not candidates for surgery, which is the preferred treatment, and, according to literature data, have a worse prognosis than operated patients [10,12,17,26,27,28,29,30]. The exclusion of the possibility of surgical treatment is more often due to the involvement of the distal pulmonary arteries (technically inaccessible), the presence of severe comorbidities (unfavorable risk-benefit ratio), or the patient’s refusal to undergo surgery. Riociguat, an oral guanylate cyclase stimulator, has been approved for patients with inoperable CTEPH or persistent/recurrent PH after PEA; other PH medications have been tested in CTEPH and are used off-label [7,27,28,29]. The off-label use of drugs approved for PH and pulmonary angioplasty can be considered in expert centers. The aim of this article is to provide a review of the diagnosis and management of CTEPH.

## 2. Epidemiology

CTEPH is a rare condition, an uncommon sequel of acute pulmonary embolism. The CTEPH incidence and prevalence are 0.9 and 3.2 cases per million, respectively [30,31,32,33,34,35,36,37,38,39,40]. The exact incidence of CTEPH has often been underestimated, and its adequate determination still remains a difficult task [33]. Prospective observational studies have reported a cumulative incidence of between 0.1–9.1% within two years after an acute pulmonary embolism event [20]. The large margin of error is due to various factors, including selection bias, the non-specificity of symptoms, and the difficulty in distinguishing acute pulmonary embolism from symptoms of pre-existing CTEPH [34,35]. Among the long-term follow-up studies carried out to determine the correct incidence of the disease, Pengo et al. reported a 3.8% incidence at a two-year follow-up in 314 patients after one episode of pulmonary embolism [34]. Another study by Becattini et al. reported an incidence of 1% in 256 patients after a 46-month follow-up period from post-acute pulmonary embolism [35]. Ribeiro et al. observed a 5.1% incidence in a group of 78 patients who had echocardiography Doppler follow-up after an episode of acute pulmonary embolism, demonstrating that the observation of an elevated pulmonary pressure by echocardiographic estimation (PAP > 50 mmHg) at the time of diagnosis of PE is associated with a high risk of persistent pulmonary hypertension [36].

However, a high percentage of patients with CTEPH do not have a history of acute pulmonary embolism. Lang et al. reported that in only about 60% of patients can a history of venous thromboembolism be confirmed [37]. According to registry data published in 2011 by Pepka-Zaba et al., only 74.8% of patients with CTEPH had a history of acute embolism, and 56.1% had a history of documented deep vein thrombosis [20].

Therefore, to date, there is little evidence in support of routine screening for CTEPH in patients with acute PE. However, although current data do not allow for the determination of the exact incidence of CTEPH, the history of pulmonary thromboembolism remains the most important risk factor for the development of this disease and the most recent guidelines for the diagnosis and management of acute PE (ERS 2019 and 2021) propose the adoption of a new algorithm for the follow-up of patients in the post-PE period and new indications as to the management of patients with the characteristics of post-embolic syndrome that includes patients other than those with CTEPH [1,38].

## 3. Pathogenesis

CTEPH differs from other pulmonary hypertension groups due to the presence of obstructive thrombotic material in the pulmonary arteries after a period of not less than three months of effective post-acute pulmonary embolism anticoagulation [38,39]. The pathophysiology of CTEPH is complex, and much remains to be clarified [10,32]. However, growing evidence suggests that CTEPH is due not only to chronic mechanical obstruction from residual thrombotic material, but also to a severe remodeling and microvasculopathy of the distal pulmonary arteries (0.1–0.5 mm) [20,40,41,42,43]. The natural history of acute PE is the resolution of the thrombus with the restoration of normal blood flow. However, although the underlying mechanism remains unclear, these emboli do not resolve in a subset of patients and remodel into fibrotic tissue, which narrows and obstructs major pulmonary arteries, eventually leading to CTEPH [40,41,42,43]. Moreover, it has been postulated that failure to resolve one or more PE episodes may lead to a significant obstruction of pulmonary blood flow (shear stress) in non-occluded areas and progressive PH due to vascular remodeling in small pulmonary vessels [34].

There are various possible triggering factors that may contribute to this sequence of events, with a consequent increased risk of thrombus persistence: infections, splenectomy, thrombophilia, clot size, and/or autoimmunity [20].

However, why some patients develop CTEPH after an episode of PE and others do not, remains to be clarified. Indeed, the thromboembolic material is replaced by granulation tissue, with the recruitment of leukocytes and endothelial cell progenitors responsible for the angiogenesis, leading to complete resolution of the thromboembolic event in most patients with PE in whom endogenous thrombolysis mechanisms have been efficacious [44,45,46]. Some studies have reported that thrombus resolution normally occurs within 30 days after an acute pulmonary embolism, and exercise tolerance is almost completely restored in most patients [44,45,46].

However, the process of resolution of the thrombus after acute pulmonary embolism is ineffective in patients with CTEPH and leads to a process of reorganization of the thromboembolic material in the vessel wall [31]. This process is characterized by a thickening of the inner layer by the formation of fibrotic tissue, with deposition of collagen and elastin fibers by alpha-SMA positive cells, potentially deriving from the differentiation of smooth muscle fiber cells that have migrated from the tunica media [47,48,49,50,51].

This reorganization of the unresolved thromboembolic material gives rise to various degrees of vascular stenosis with the formation of ring, network, or band-like structures that obstruct blood flow, leading to an increase in vascular resistance and pulmonary arterial pressure. The underlying mechanisms responsible for ineffective thrombolysis and the persistence of thromboembolic material in the pulmonary arteries of CTEPH patients have not yet been identified [20]. Numerous literature reports suggest that inflammation plays a role in the pathogenesis of CTEPH. This hypothesis is supported by the high prevalence of inflammatory diseases in CTEPH patients who have increased plasma levels of some cytokines and interleukins (IL-1beta, IL-4, IL-8, and IL-10) and elevated plasma levels of TNF- alpha [50,51,52]. Indeed, Quark et al. assessed the hypothesis that chronic inflammation and ineffective angiogenesis are involved in the pathogenesis of CTEPH [50]. They reported that some inflammation markers, i.e., C-Reactive Protein (CRP), IL-10, Chemotactic Monocyte Protein (MCP-1), the Macrophage Inflammatory Protein (MIP1-1alpha), and Matrix Metalloproteinases (MMP-9), were higher in CTEPH patients than in healthy controls. Moreover, they reported observing a reduced angiogenic gene expression and fewer SMA positive smooth muscle cells in vascular thromboembolic material removed by PEA in both non-surviving CTEPH patients and in those with persistent pulmonary hypertension after surgery. Their data support that reduced angiogenesis may be correlated with increased mortality or disease progression [52].

The obstacle to the pulmonary blood flow due to the fibrotic thromboembolic material leads to an increase in resistance and pressure in the non-occluded areas, with consequent significant ‘shear stress’. This promotes the remodeling process in the wall of the small distal vessels and the progressive development of pulmonary hypertension [31]. Varying types of small vessel pathology have been observed in CTEPH patients and those with distal perfusion defects who are not candidates for PEA surgery [51,52,53]. These include obstruction of the small elastic subsegmental arteries, thickening of the intima, hypertrophy of the media with plexiform lesions in the small muscular arteries and arterioles, which, on histological analysis, can be likened to the alterations observed in the vascular lesions of patients with idiopathic pulmonary arterial hypertension (IPAH) [31]. Therefore, the pathology of the distal pulmonary vessels contributes to CTEPH progression and may well be responsible for the persistence of pulmonary hypertension in about 10% of patients where the thromboembolic material has been surgically removed by pulmonary endarterectomy [50,51,52,53].

## 4. Risk Factors

The risk factors associated with the onset of CTEPH reported in the literature include aspects related to the acute embolic event, the presence of some conditions associated with thrombophilia, and some concomitant pathological conditions [53,54,55,56,57,58]. Pengo et al. identified the presence of previous episodes of PE or deep vein thrombosis (DVT) as risk factors for CTEPH, they are also reported that the greater extent of the initial embolic perfusion defect and the idiopathic presentation of pulmonary embolism are involved in the CTEPH process [34]. Clinical conditions reported as risk factors for the development of CTEPH include ventricle-atrium shunt for the treatment of hydrocephalus, splenectomy, inflammatory bowel disease, osteomyelitis, myeloproliferative syndromes, the presence of infected pacemakers/venous catheters, and hypothyroidism in hormone replacement therapy as it is associated with increased plasma levels of the Von Willebrand factor and a history of neoplastic pathology [32,41,55,56,57,58].

Some studies carried out in three European centers evaluated the incidence of thrombophilic risk factors commonly associated with venous thromboembolism. They reported that the presence of Lupus Anticoagulant (LAC)/Antiphospholipid Antibodies was a risk factor, as were elevated plasma factor VIII levels, in 10–20% and 25% of patients, respectively [41,59,60,61,62]. No correlation was observed between deficiencies in antithrombin III, protein C, protein S, or the factor V Leiden mutation with a predisposition to developing CTEPH [20,51,52,53,59].

## 5. Clinical Presentation

CTEPH occurs mainly in adulthood, at an average age of 63 years, with no significant difference between gender [32]. Early identification and an accurate diagnosis of CTEPH are difficult and take an average of 14 months from the onset of symptoms and diagnosis in expert centers [43]. Diagnostic difficulties are determined by the varying characteristics of the pathology, which include the patient being completely asymptomatic during the development period (the so-called *honeymoon period*) after an embolic event and clinical signs and symptoms which are largely nonspecific and related to progressive right heart failure [31,32].

The clinical signs and symptoms at diagnosis are similar to those of patients affected by IPAH and, according to the data reported in the international registry of patients with CTEPH, they include dyspnea (99.1%), declining edema (40.5%), fatigue and asthenia (31.5%), chest pain (15.3%) and syncope (13.7%) [10,31,32]. Syncope typically occurs in patients with IPAH, while haemoptoe is a more common observation in CTEPH due to the presence of bronchial artery hypertrophy, which in turn is caused by reperfusion of the ischemic lung region by the bronchial arterial circulation [32,61]. As the right ventricular function declines, there is a gradual appearance of high jugular venous pressure, hepatosplenomegaly, ascites, and peripheral edema [39].

## 6. Diagnosis

The timely diagnosis of CTEPH, followed by referral to a specialized center, is important for choosing the right treatment options. The diagnosis of CTEPH is based on the observation of precapillary pulmonary hypertension (defined as a mean pulmonary arterial pressure (mPAP) ≥ 25 mmHg with a capillary pulmonary pressure ≤ 15 mmHg) assessed by right cardiac catheterization, associated with at least one segmental perfusion defect evident on ventilation scan/perfusion and signs of chronic thromboembolism on chest CT angiography or conventional pulmonary angiography (ring stenosis, chronic lesions and/or complete occlusions), in patients who had effective anticoagulation therapy for at least three months after a pulmonary embolism episode [38,39]. The application of the diagnostic algorithm provides an anamnestic evaluation of the signs and symptoms suggestive of CTEPH, followed by a radiological evaluation and lung ventilation-perfusion scintigraphy, all of which is to be confirmed by a hemodynamic evaluation [1,38,60,61,62,63,64].

Although CT pulmonary angiography is the first choice for diagnosing acute pulmonary embolism, ventilation-perfusion lung scintigraphy (V/PSCAN) remains the first-line imaging technique for diagnosing CTEPH, as it has a sensitivity of 96–97% and a specificity of 90–95% [63]. A standard V/PSCA lung scan allows for a confirmation/exclusion of a diagnosis of CTEPH. Although pulmonary angio-CT is a recognized imaging technique, when used alone it cannot rule out a diagnosis of CTEPH [63,64]. Pulmonary angio-CT is able to evidence the typical alterations, comorbidities, and complications observed in CTEPH, e.g., dilation of the pulmonary arteries or large bronchial arterial collaterals, as well as areas of hypoperfusion (hyper-transparency) that characterize the “mosaic” pattern (Figure 1). Although commonly observed in CTEPH patients, the mosaic pattern may also be present in 12% of patients with IPAH [65]. The chest HRTC is also able to provide important data on the lung parenchyma, which may be affected by alterations due to interstitial diseases, pulmonary emphysema, or bronchial pathologies.

Pulmonary angiography is the final step in confirming a diagnosis of CTEPH and supports the technical evaluation for surgery eligibility. The typical signs observed in pulmonary angiography are ring stenosis, reticular lesions, tortuous lesions, and complete vascular obstructions. MRI is also able to evidence the typical signs of CTEPH, such as pulmonary vessel stenosis, reticula, and occlusions, and can be a complementary test in the diagnosis of CTEPH [39,66].

Right cardiac catheterization is fundamental for diagnosis. This test assesses the pressure in the right atrium, right ventricle, and pulmonary artery, determines the right cardiac output and the saturation of mixed venous blood, and calculates the pulmonary vascular resistance, which is an important long-term prognostic factor in the preoperative and postoperative period (Figure 2) [67,68].

A correct differential diagnosis between CTEPH and IPAH is essential to make a therapeutic decision, as CTEPH can potentially be resolved by endarterectomy. Care is to be taken to distinguish CTEPH from other pathologies that can mimic the radiological picture of CTEPH, i.e., pulmonary artery sarcoma, tumor cell embolism, mediastinal fibrosis, arteritis of the great vessels (Takayasu’s arteritis), parasites such as hydatid cysts, foreign body embolism, and congenital or acquired pulmonary vascular stenosis [69].

## 7. The “Post Embolic Syndrome”

Cohort studies carried out over the past few decades have reported that, although CTEPH is considered a rare disease, persistent or worsening dyspnea and impaired exercise capacity are frequently present six months to three years after an episode of acute PE [70]. The percentage of patients who reported a deterioration in their health at a six-month follow-up after pulmonary embolism diagnosis was highly variable, ranging from 20% to 75% [37,38,61].

More recently, a prospective cohort study carried out from 2010 and 2013 in five Canadian hospitals on 100 patients with a one-year follow-up after a pulmonary embolism episode, reported that 47% of these patients developed impaired maximal aerobic capacity, defined as a reduction in maximal predicted oxygen uptake to <80%, at the cardiopulmonary exercise test. This functional outcome was associated with a significant reduction in dyspnea and quality of life scores and, above all, in the distance covered at the Six Minute Walking Test [71]. Independent predictors for the development of impaired exercise capacity and a reduction in the quality of life after a pulmonary embolism event were: the female gender, an elevated BMI, the presence of a respiratory pathology, a high pulmonary arterial pressure estimated by echocardiogram 10 days after the diagnosis of pulmonary embolism, and the largest average pulmonary artery diameter on chest CT angiography [72]. However, the pulmonary function tests and echocardiogram performed during the follow-up were normal in the vast majority of patients, with or without reduced maximal aerobic capacity [71]. A study carried out on 20 survivors of an episode of massive/sub-massive pulmonary embolism did not demonstrate any association between the onset of a reduced physical exercise capacity and the persistence of right ventricular dilation/dysfunction on echocardiographic evaluation [73,74].

On the whole, these studies suggest that muscle deconditioning, mainly affecting subjects with a high BMI and cardio-respiratory diseases, is largely responsible for the dyspnea and the signs of functional limitation frequently observed in patients after an acute pulmonary embolism. However, at least in most cases, the reduced exercise capacity after pulmonary embolism does not seem to be attributable to the presence of extensive perfusion defects or persistent/progressive pulmonary hypertension or right ventricular dysfunction [44].

The expression “post-embolic syndrome” has been used to describe persistent dyspnea, reduced exercise tolerance, and reduced quality of life that persist for more than three months after a period of effective post-acute pulmonary embolism anticoagulation. This patient group may include patients with pulmonary hypertension (CTEPH), those with evidence of perfusion defects without pulmonary hypertension (CTED), and symptomatic patients without evidence of pulmonary vascular disease (dyspnea associated with post-embolism) [75]. Patients with chronic thromboembolic disease (CTED) have a clinical and radiological picture similar to that of patients with CTEPH, with a hemodynamic picture (assessed by right cardiac catheterization) which excludes the presence of pulmonary hypertension [33].

Patients with CTED generally have a functional picture of limited physical exercise capacity during cardiopulmonary exercise testing (CPET), with characteristics that point to the presence of vascular disease, e.g., increased dead space ventilation. However, the exercise capacity of these patients was greater than that of patients with CTEPH, and patients with CTED generally do not develop right heart failure [75]. The hemodynamic picture of patients with CTED during right cardiac catheterization at rest shows normal/borderline pulmonary vascular pressure and resistance values for pulmonary hypertension (mPAP 20–24 mmHg, PVR 2–3 Wood units). Cardiac catheterization performed during physical exercise shows an increase in the mPAP/CO ratio and a marked increase in mPAP (>30–35 mmHg), not accompanied by an adequate fall in pulmonary vascular resistance [74].

New values were recently proposed for the definition of precapillary pulmonary hypertension (mPAP > 20 mmHg, PAW </= 15 mmHg and PVR >/= 3 Woods units) during the 6th World Symposium on Pulmonary Hypertension (WSPH). Although there is evidence that suggests the introduction of these threshold values into clinical practice, the consequences on the diagnosis of CTEPH and CTED, respectively, have not yet been established [1,75,76]. As aforementioned, the incidence of CTED is still under debate, while historical data indicate that progression from CTED to CTEPH is an uncommon observation. Therefore, there is a paucity of current evidence able to guide the management and treatment of these patients [75].

Although there is little evidence in support of the adoption of a screening program in asymptomatic patients after an episode of acute pulmonary embolism, it has been reported that an in-depth diagnostic study in patients with persistent symptoms leads to a diagnosis of CTEPH/CTED in 8.5% of cases [76]. The ESC/ERS 2019 guidelines for the diagnosis and management of acute pulmonary embolism have proposed a follow-up strategy for patients who survive an episode of acute pulmonary embolism after hospital discharge. This strategy provides for the follow-up of all patients after three to six months, aimed at assessing the clinical-functional picture and risk factors associated with CTEPH. In patients who report dyspnea and/or reduced exercise tolerance, an echocardiographic examination is recommended as the next step to assess the probability of chronic pulmonary hypertension and, therefore, of CTEPH [1,38]. The 2015 ESC/ERS guidelines for the diagnosis and treatment of pulmonary hypertension also recommend echocardiography in patients with symptoms suggestive of CTEPH. The echocardiography should suggest a high or intermediate probability of pulmonary hypertension, then lung perfusion scintigraphy is recommended [39].

In the light of the fact that the echocardiographic examination may underestimate a diagnosis of CTEPH or CTED, at least in patients with pronounced symptoms, it is inappropriate to interrupt the diagnostic path if the echocardiogram shows a low probability of pulmonary hypertension and it is too simplistic to start a follow-up with an echocardiogram at three to six months (40). As patients with CTEPH and CTED may show signs of pulmonary perfusion abnormalities at the cardiopulmonary exercise test (CPET), it is advisable to perform this test when clinical suspicion of CTEPH/CTED persists with a low probability of pulmonary hypertension on the echocardiogram. A hemodynamic evaluation with right cardiac catheterization, a radiological study and scintigraphy, and, when appropriate, the cardio-pulmonary exercise test are recommended when there is a strong clinical suspicion of PH after the echocardiogram [67].

It has been reported that, in some selected cases, patients with CTED can benefit from surgical treatment of pulmonary endarterectomy with an improvement of the hemodynamic picture, functional class, and physical exercise capacity (12). Pulmonary balloon angioplasty has also been performed in selected CTED patients, with evidence of hemodynamic and functional capacity improvement [77]. Currently, the natural history of CTED remains unknown, and there is no evidence to guide the management and treatment of this group of patients in need of symptom relief interventions. The guidelines for the treatment of patients with CTEPH cannot be applied to patients with CTED [33]. While for patients with CTEPH, lifetime anticoagulation is recommended for patients with CTED.

## 8. Treatment Options

### 8.1. Surgical Treatment

As aforementioned, CTEPH is a potentially treatable disease, and the treatment of choice is pulmonary endarterectomy (Figure 3) [12,13,38,39].

This procedure involves a bilateral endarterectomy of the pulmonary arteries with the removal of the obstructive thromboembolic material from the vessel wall. The surgical approach requires a median sternotomy and the use of a heart-lung machine with deep hypothermia and circulatory arrest. In some cases of severe postoperative pulmonary reperfusion edema, the use of veno-arterial or veno-venous extracorporeal membrane oxygenation (ECMO) may be required [66,67,76,77,78].

The hemodynamic result after PEA is drastic and immediately evident in the operating room, with a return in most cases to normal values. Within the first few days after surgery, there is also an initial recovery of cardiac function, with a clear reduction of right ventricular dilatation. After one year, there is a reverse remodeling of the right ventricle, with the disappearance of ventricular hypertrophy and the recovery of a normal ventricular wall motion. The functional improvement of patients, instead, occurs more gradually and is linked to the patient’s age or to the duration of the disease [79,80,81,82].

The hospital mortality of PEA depends on the degree of experience of the center, evaluated in terms of annual interventions performed. It is about 3.5% in specialized centers that perform at least 50 interventions/year, 4.7% in centers with 11–50 interventions/year, and 7.4% in centers that perform less than 11 operations/year [67]. The most important predictor of hospital mortality is the patient’s preoperative hemodynamic situation: in a large registry of 2700 CTEPH patients, a PVR value higher than 1000 dynes/s/cm (-5) led to an overall mortality of 4.1%, while in patients with PVR values less than 1000 dynes/s/cm (-5) mortality was 1.6% [81,82]. This is in agreement with the observation of Dartevelle et al., reporting an overall mortality of 4% in patients with PVR 900 dynes/s/cm (-5), which increased by 10% in patients with a PVR between 900–1200 dynes/s/cm (-5) [79].

Jamieson’s anatomical classification distinguishes four types of thromboembolic lesions based on the characteristics of the pathological tissue samples obtained during pulmonary endarterectomy. Type I: lesions localized at the level of the major proximal vessels, consisting of fresh thromboembolic material, in the organization phase and evident immediately after the surgical incision of the pulmonary artery. Type II: lesions characteristic of chronic vascular pathology in the proximal pulmonary arteries, with intimal thickening and without fresh thrombotic material. Type III: lesions characterized by a more distal location, with the involvement of the segmental and subsegmental arteries, leading to greater difficulty during surgery and the need for the surgeon to find the dissection plane in each of the segmental and subsegmental arteries. Type IV: this type affects a very small subgroup of patients in whom, despite a complete surgical exploration and intimectomy of the vascular tree, the lesions do not show evidence of thromboembolic material [77,83].

The San Diego Hospital Study reported that types II and III lesions are the most common, with an incidence of 38% and 39% of cases, respectively. The authors observed a significant reduction of average PVR and PAP values after the endarterectomy. A total of 1410 patients had post-surgical follow-up and the survival rate was 82% at 5 years and 75% at 10 [81,82,83].

The International Registry included 679 CTEPH patients. Endarterectomy was performed in 386 patients and significantly improved the patients’ hemodynamic picture, with a reduction in the average PVR value from 698 dynes/s/cm (-5) to 235 dynes/s/cm (-5) one year after surgery. The hospital mortality rate of the operated patients was 4.7% [32]. In the International Registry, although operable patients were younger than inoperable patients, they showed no differences in symptoms, NYHA functional class, and hemodynamic picture. However, the two groups had a different frequency of some associated pathological conditions. In particular, thrombophilic disorders were more frequent in operable patients, while splenectomy and cancer were more frequent in inoperable patients. These differences indicate that the two groups, i.e., operable and inoperable subjects, may represent two distinct subpopulations of CTEPH patients [10].

Although criteria for the operability of patients have been proposed, to date, the surgical decision is influenced by the experience of the reference center for the intervention. Indeed, the experience of the surgical and multidisciplinary team is pivotal in determining which vascular lesions are to be considered proximal and therefore surgically accessible [79,80,81,82]. Even if the combination of a predominantly distal obstruction and high PVR values is a risk factor, it is not a contraindication for PEA. In these patients, although peripheral microangiopathy is prevalent, PEA can be useful to reduce PVR and consequently the overload on the right ventricle. Even advanced age is not an exclusion criterion for the intervention. Based on this evidence, an evaluation of endarterectomy eligibility in an expert center is recommended for all CTEPH patients [67].

### 8.2. Pharmacological Treatment

Three trials in CTEPH—the Phase III CHEST-1 trial (riociguat), the Phase III CTREPH trial (treprostinil), and the Phase II MERIT-1 trial (macitentan)—reached their primary endpoints.

No head-to-head trials have been carried out with riociguat versus other medical therapies for the treatment of CTEPH, and differences in study design limit direct comparisons between studies. The main differences between CHEST1, CTREPH, and MERIT-1 are that MERIT-1 only enrolled patients with inoperable CTEPH, whereas CHEST-1 and CTREPH included patients with inoperable CTEPH and patients with persistent/recurrent CTEPH [1]. Moreover, MERIT-1 and CTREPH allowed PAH therapy at baseline, whereas CHEST-1 included treatment-naïve patients only [1,69,70,71]. Lastly, the patients enrolled into the CTREPH were older overall and had more severe disease than those in the CHEST-1 and MERIT-1 studies [1,69,70,71].

Medical therapy and balloon pulmonary angioplasty (BPA) have provided alternate therapeutic options for patients with inoperable CTEPH, although there is a paucity of literature on the outcomes. Pharmacological treatment of CTEPH includes lifetime anticoagulant therapy and additional drugs aimed at treating symptoms and congestive heart failure, e.g., diuretics, oxygen, when required. Lifetime anticoagulant therapy is strongly recommended, even after PEA surgery, while there are no data on the efficacy and safety of the new oral anticoagulants compared to vitamin K inhibitors [1,39]. The goal of anticoagulant therapy in CTEPH is the prevention of thrombosis in situ of the pulmonary arteries and episodes of venous thromboembolism. Positive effects of gradual and controlled re-training to physical exercise have been demonstrated in patients with CTEPH [67].

The rationale for using antihypertensive drugs approved in IPAH is based on the treatment of pulmonary microangiopathy in patients with CTEPH who are not eligible for surgery and in patients with relapse of PH post-PEA [1,39]. Based on data from an international registry, 36% of patients were deemed inoperable due to distal thromboembolic lesions or the presence of severe co-morbidities. In several studies, the persistence of an elevated mPAP after surgery was estimated to vary greatly, from 5% to 35% [32,81,82,83,84]. According to the same register, 37.9% of patients started at least one therapy for pulmonary hypertension at the time of diagnosis (28.3% of operable patients and 53.8% of inoperable patients) [32,83,84,85,86,87].

Although several randomized controlled studies have been carried out to evaluate the efficacy and safety of this therapeutic option, currently, the only drug registered for pulmonary hypertension and also approved by the FDA for CTEPH is riociguat. Riociguat is a stimulator of Guanylate Cyclase (sGC) (IB recommendations) for the management of inoperable CTEPH and persistent/recurrent CTEPH after PEA [21]. It has a dual-mode of action within the NO–sGC–cGMP pathway, stimulating sGC directly via a NO-independent binding site and stabilizing the binding of NO to sGC [14]. Other vasodilator drugs approved for pulmonary hypertension are off-label treatments in CTEPH and their prescription should be reserved to expert centers (Recommendation IIb B) [1,39].

In the Chronic Thromboembolic Pulmonary Hypertension Soluble Guanylate Cyclase-Stimulator Trial 1 (CHEST-1) study, it was demonstrated that riociguat significantly improved the results of the 6-min walking distance test (6MWD) (average +39 +/− 79 m) and the World Health Organization (WHO) functional class and decreased PVR and N-terminal pro-brain natriuretic peptide (NT-proBNP) levels in 261 patients at 16 weeks after initiation of treatment compared to placebo. Therefore, it was approved for the treatment of CTEPH in inoperable patients with persistent PH after PEA. The long-term extension study (CHEST-2) demonstrated that the use of riociguat is safe and efficacious up to one year after treatment initiation. However, there is limited availability of long-term follow-up data and experiences from the ‘real world’ [1,39,85,86,87,88,89].

Endothelin-1 is a powerful vasoconstrictor and mitogenic factor of smooth muscle cells synthesized and secreted by endothelial cells. Plasma levels of endothelin are elevated in both patients with IPAH and those with CTEPH [80]. Endothelin receptor antagonists (ERAs) act selectively by blocking type A receptors or non-selectively by blocking both type A and type B receptors, thus preventing the vasoconstrictor and mitogenic response to endothelin-1 [85,86,87,88,89]. In the BENEFIT study, bosentan (dual endothelin inhibitor) achieved only one of the two primary endpoints, i.e., it was effective in reducing PVR but did not show significant effects on the distance covered during the 6MWD in 77 patients with symptomatic CTEPH after 16 weeks of treatment [90,91,92]. However, the BENEFIT study reported no statistically significant difference in the time to clinical deterioration in the bosentan-treated patients compared to the placebo group (80 patients) [90].

A recent study on the efficacy of macitentan, a dual endothelin receptor antagonist, on 80 patients with unresectable CTEPH (MERIT-1), reported a reduction in PVR to 73% from baseline (primary endpoint) and an increase in 6MWD (+35 m) at the 24-week assessment after starting treatment. However, this study only included inoperable patients and not those with persistent PH post-PEA, and also allowed the use of a combination therapy of macitentan with other drugs for pulmonary hypertension in 64% of patients [67].

Prostacyclin is a powerful vasodilator produced by endothelial cells that inhibits platelet aggregation and the proliferation of smooth muscle cells [93]. As prostacyclin plasma levels are reduced in IPAH patients, prostanoid drugs have been used in CTEPH patients [85,86,93]. Intravenous epoprostenol was administered in 27 patients with inoperable CTEPH, with a significant increase in physical exercise capacity and Cardiac Index (CI) at a 20-month follow-up. However, further studies are required as to the efficacy of these drugs in CTEPH [94].

Nitric oxide (NO), an endogenous vasodilator produced by endothelial cells, inhibits platelet aggregation and smooth muscle cell growth, by activating soluble guanylate cyclase (sGC) for cGMP synthesis, which leads to smooth muscle relaxation [95].

Sildenafil, a Phosphodiesterase-5 inhibitor, works by preventing the degradation of cGMP, and its efficacy was evaluated in a pilot study on 19 patients with inoperable CTEPH or persistent PH after surgery (Sildenafil Study) [96]. The study reported that although sildenafil therapy does not significantly increase the 6MWD at three months (primary endpoint), it does lead to a higher reduction in PVR and an improvement in the WHO functional class (secondary endpoints) compared to the placebo group [96].

Overall, three large controlled trials, i.e., BENEFIT, CHEST, and MERIT, along with a large number of small studies published in the literature, have shown that drugs used in the treatment of pulmonary hypertension are able to induce various improvements in the hemodynamic and functional pictures and, on the basis of the evidence provided, the 2015 ESC/ERS Guidelines justified their use in patients with inoperable CTEPH and in those with persistent PH post-PEA. However, the optimal use and benefits of many therapies are still unclear and there was often a poor correlation between hemodynamic and functional outcomes, emphasizing the need for further controlled trials. The challenges in the future development of medical therapy for CTEPH are therefore represented by a greater understanding of the disease, new study designs, and endpoints to monitor the progression of the disease, the optimization of pulmonary hypertension therapies tailored to the different patient characteristics, and emerging treatment options, such as balloon pulmonary angioplasty [80].

Preoperative treatment (bridge therapy) with approved vasodilator drugs for pulmonary hypertension is rather controversial as it is associated with a delay in carrying out surgery. In the CTEPH International Patient Registry and the San Diego Hospital Study, 28% and 37% of the operated patients, respectively, had received medical treatment for pulmonary hypertension prior to surgery, and the time interval between diagnosis and endarterectomy was doubled, with no clinical benefits in either group [32,81]. Reesnik et al. reported a benefit in the clinical and hemodynamic pictures due to treatment with bosentan before endarterectomy. However, the number of trials carried out to date does not suffice to demonstrate the advantage of this preoperative medical treatment [97,98].

### 8.3. Pulmonary Balloon Angioplasty

Pulmonary balloon angioplasty (BPA) is a promising alternative therapy for patients with inoperable CTEPH and has been reported as a therapeutic option in highly selected patients (Recommendation IIb C) [1,39]. This procedure involves dilating and unblocking the vessel wall of the stenosing material with a balloon catheter. Candidate patients are those with inoperable CTEPH who have catheter-accessible thrombotic lesions. These lesions are generally nets and fissures, while lesions that completely occlude the pulmonary arteries cannot be treated by BPA. Whether or not BPA is an option for a determined patient must be decided according to the operator’s experience and lesions’ type, distribution, and location. There are currently no reported cases of stenosis after BPA [1,99,100].

This technique, currently mainly used in Japan, is also being developed in European expert centers. An average of 4.8 sessions are needed to obtain an improvement in a patient’s hemodynamic picture. The literature describes significant improvements in the hemodynamic picture of patients undergoing BPA [99,100,101,102,103,104,105]. Indeed, in a study by Mizoguchi et al., 69 patients (94% of treated patients) were in functional class I or II after BPA and a significant reduction in the average pulmonary arterial pressure from 45 (+/−9.6) to 24 (+/−6.4) mmHg (*p* < 0.01) was observed. BPA-associated mortality was 0–3.4%, while BPA-related lung injury was 9–60% of cases [99,104]. However, this procedure is not without complications, and lung reperfusion injury is the main one. Therefore, some centers take measures to reduce reperfusion injury during BPA by performing it in separate and limited sessions to two segments of the same lung until mPAP falls below 30 mmHg. A higher PAP at BPA has been observed to be associated with a higher frequency of post-treatment lung injury [100]. Moreover, there is also a potential risk that the guidewire may perforate the pulmonary artery or rupture vessels. Consequently, to reduce the risk of damage to the vascular walls, dilation of the lesions should be as small as is feasibly possible [101].

## 9. Conclusions

There is a high risk of death for right heart failure in undiagnosed/untreated CTEPH patients. The scientific community has provided major contributions to CTEPH over the last few decades and several recent studies provided a better understanding of the diagnosis and treatment of this disease. Following the 6th World Symposium on PH, pulmonary endarterectomy is the preferred treatment of choice for operable CTEPH patients. Targeted medical therapy and balloon pulmonary angioplasty are promising alternative treatment options in inoperable patients. Indeed, patients undergoing surgical treatment had an 80% reduction of their PVR value and a five-year survival of 90%, while patients treated with medical therapy had a 25% reduction in PVR and a three-year survival of 70%.

In conclusion, CTEPH is a complex disease that requires an integrated, multidisciplinary approach with a dedicated team to diagnose, treat, and follow up on CTEPH patients.

## Figures and Tables

**Figure 1 diagnostics-12-00235-f001:**
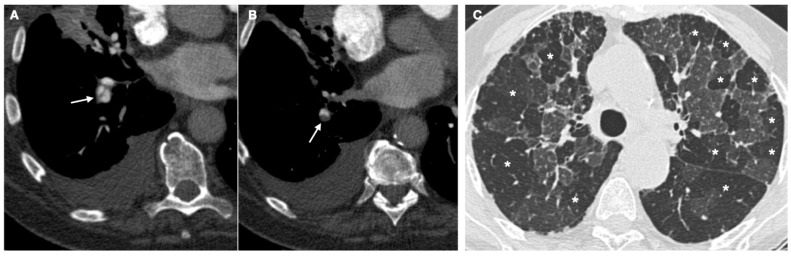
(**A**) Axial images from a pulmonary angio-CT demonstrating the presence of bands within the vascular lumen (arrow) and (**B**) a focal vascular stenosis secondary to an eccentrically located thrombus (arrow), due to incomplete thrombus resolution. (**C**) HRTC showing a diffuse mosaic pattern of perfusion (*), a typical radiological finding in chronic pulmonary thromboembolism.

**Figure 2 diagnostics-12-00235-f002:**
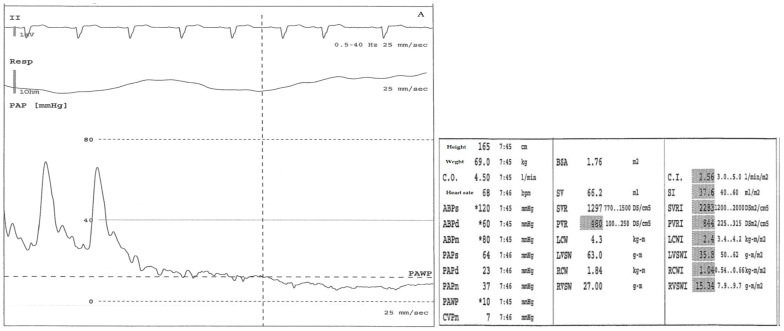
(**A**) Right cardiac catheterization in a CTEPH patient, not a candidate for surgical treatment due to the involvement of the distal pulmonary arteries (technically inaccessible), before treatment (as reported in the table at right PVR 480 dynes/s/cm (-5)), * software evaluation. From below: pulmonary arterial pressure, respiratory, and ECG waveforms during arterial catheterization. The first part of the pressure trace reflects the pressure in a pulmonary artery (large swings, dicrotic notch). Then, the balloon is inflated, and the tip of the Swan Ganz catheter floats until it wedges in a small artery (small swings synchronous with respiratory rate). This provides a pulmonary arterial wedge pressure (PAWP), i.e., an indirect measure of the pressure in the left ventricle. (**B**) Right cardiac catheterization in the same patient after three months of treatment with riociguat as per data sheet (as reported in the table at right PVR 418 dynes/s/cm (-5)).

**Figure 3 diagnostics-12-00235-f003:**
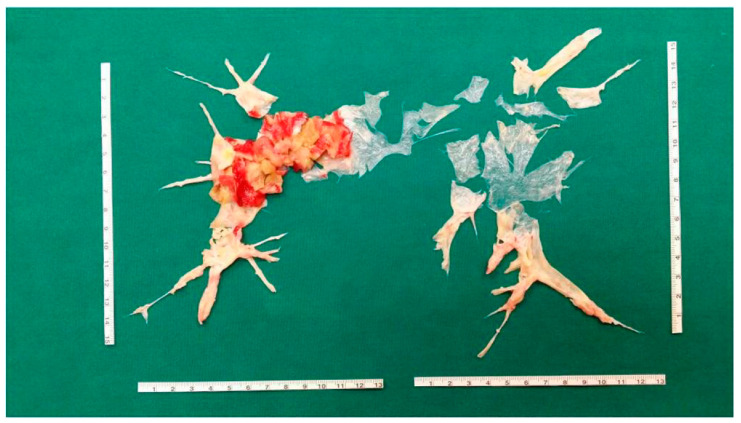
Pulmonary endarterectomy casts from a patient with complex and severe CTEPH.

## Data Availability

Not applicable.

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
