# Peer review of "Chronic Thromboembolic Pulmonary Hypertension: An Update"

_diagnostics, 2022, doi:10.3390/diagnostics12020235_

Round 1

Reviewer 1 Report

I was pleased to review your manuscript. I make the following minor suggestions": - when citing Lang et al in a sentence the phrase 'in about 60% of patients' is unnecessarily repeated. - In Section 5 - Clinical Presentation, the phrase "clinical sign and symptoms which are largely nonspecific" should be corrected to "clinical signs and symptoms which are largely nonspecific" - The spelling of 'availability' should be corrected in the sentence "However, there is limited avialability of long-term follow-up data 521 and experiences..."

Author Response

I was pleased to review your manuscript. I make the minor suggestions.

R: We would like to thank the reviewer for the comments

1)When citing Lang et al in a sentence the phrase 'in about 60% of patients' is unnecessarily repeated.

R: In agreement with the reviewer, we delete the repetition.

2) In Section 5 - Clinical Presentation, the phrase "clinical sign and symptoms which are largely nonspecific" should be corrected to "clinical signs and symptoms which are largely nonspecific"

R: In agreement with the reviewer, we correct the phrase.

3) The spelling of 'availability' should be corrected in the sentence "However, there is limited avialability of long-term follow-up data 521 and experiences..."

R: In agreement with the reviewer, we correct the spelling.

Reviewer 2 Report

This paper provides an overview of chronic thromboembolic pulmonary hypertension (CTEPH). The overview includes epidemiology, pathogenesis, risk factors, and treatment options. A special chapter, "The post embolic syndrome," addresses chronic thromboembolic disease and its differences from CTEPH. The review is comprehensive, and the literature is up to date. However, I would like to make a few points for the authors to consider:

  1. lines 77-78: the authors state that a considerable number of CTEPH patients are not eligible for surgery. This statement is somehow confusing, vague and can be misinterpreted.The same is repeated in lines 483-484. The treatment of choice for CTEPH is surgery, and a significant number of patients are operable. Please change "significant" to "certain" and indicate the exact percentage of operable patients according to the different studies.
  2. Line 87: Please use bold letters for "2. Epidemiology".
  3. Line 206: Please use bold letters for “5. Clinical presentation”.
  4. in lines 220-221, please use "development of bronchial collaterals" for ischemic lung regions. And please delete "e cosi" at the end of the sentence.
  5. line 247-248: evidence on CT of "large bronchial arterial collaterals" instead of "bleeding of collateral bronchial arteries".
  6. line 251: Please provide the abbreviation for "HRTC".
  7. line 402-406: Please reword the entire sentence for "Surgical treatment". I would suggest the following sentence "The surgical approach requires a median sternotomy and the use of a heart-lung machine with deep hypothermia and circulatory arrest. In some cases of severe postoperative pulmonary reperfusion edema, the use of veno-arterial or veno-venous extracorporeal membrane oxygenation (ECMO) may be required." Reference 69 is not appropriate, please change it.
  8. line 497-499: The sentence is a bit confusing. Please provide studies to indicate in how many patients (percentage) the elevated mPAP levels persist after surgery.
  9. lines 613-616: The statements regarding surgical treatment are somewhat confusing. (1) Not only following the 6th World Symposium on PH, but also according to all previous studies. (2) "Pulmonary endarterectomy remains the preferred treatment of choice..." Please change "remains" with "is".  (3) "Targeted drug therapy and balloon angioplasty are promising alternative treatment options" Please add "in inoperable patients."
  10. Please remove all references from the "Conclusions" chapter and reword the entire chapter. The conclusions should be a concluding sequence based on the text of the manuscript.
  11. Please check the spelling of words and sentences of the whole manuscript by a native speaker.
  12. Considering that the main treatment of CTEPH is surgery, I would suggest to include a surgeon in the author list who is an expert in PEAs and who could write the chapter on surgical treatment in more detail and provide some intraoperative images to make the manuscript more representative.

Author Response

This paper provides an overview of chronic thromboembolic pulmonary hypertension (CTEPH). The overview includes epidemiology, pathogenesis, risk factors, and treatment options. A special chapter, "The post embolic syndrome," addresses chronic thromboembolic disease and its differences from CTEPH. The review is comprehensive, and the literature is up to date. However, I would like to make a few points for the authors to consider:

R: We would like to thank the reviewer for the comments

  1. Lines 77-78: the authors state that a considerable number of CTEPH patients are not eligible for surgery. This statement is somehow confusing, vague and can be misinterpreted. The same is repeated in lines 483-484. The treatment of choice for CTEPH is surgery, and a significant number of patients are operable. Please change "significant" to "certain" and indicate the exact percentage of operable patients according to the different studies.

R: In agreement with the reviewer, we correct the phrase.

  1. Line 87: Please use bold letters for "2. Epidemiology".

R: In agreement with the reviewer, we use the bold letters.

3.Line 206: Please use bold letters for “5. Clinical presentation”.

R: In agreement with the reviewer, we use the bold letters.

  1. In lines 220-221, please use "development of bronchial collaterals" for ischemic lung regions. And please delete "e cosi" at the end of the sentence.

R: In agreement with the reviewer, we delete the two words

  1. Line 247-248: evidence on CT of "large bronchial arterial collaterals" instead of "bleeding of collateral bronchial arteries".

R: In agreement with the reviewer, we correct the sentence.

  1. Line 251: Please provide the abbreviation for "HRTC".

R: In agreement with the reviewer, we correct the abbreviation

  1. line 402-406: Please reword the entire sentence for "Surgical treatment". I would suggest the following sentence "The surgical approach requires a median sternotomy and the use of a heart-lung machine with deep hypothermia and circulatory arrest. In some cases of severe postoperative pulmonary reperfusion edema, the use of veno-arterial or veno-venous extracorporeal membrane oxygenation (ECMO) may be required." Reference 69 is not appropriate, please change it.

R: In agreement with reviewer, we reword the sentence and correct the reference.

  1. Line 497-499: The sentence is a bit confusing. Please provide studies to indicate in how many patients (percentage) the elevated mPAP levels persist after surgery.

R: In agreement with reviewer, we reword the sentence and correct the reference.

  1. Lines 613-616: The statements regarding surgical treatment are somewhat confusing. (1) Not only following the 6th World Symposium on PH, but also according to all previous studies. (2) "Pulmonary endarterectomy remains the preferred treatment of choice..." Please change "remains" with "is".  (3) "Targeted drug therapy and balloon angioplasty are promising alternative treatment options" Please add "in inoperable patients."

R: In agreement with reviewer, we reword the sentences.

  1. Please remove all references from the "Conclusions" chapter and reword the entire chapter. The conclusions should be a concluding sequence based on the text of the manuscript.

R: In agreement with reviewer, we remove all references from the "Conclusions" chapter and reword the entire chapter.

  1. Please check the spelling of words and sentences of the whole manuscript by a native speaker.

R: In agreement with the reviewer, English was revised

  1. Considering that the main treatment of CTEPH is surgery, I would suggest to include a surgeon in the author list who is an expert in PEAs and who could write the chapter on surgical treatment in more detail and provide some intraoperative images to make the manuscript more representative.

R: In agreement with the reviewer, we include a surgeon in the author list who is an expert in PEAs. He write the chapter on surgical treatment in more detail and provide some intraoperative images to make the manuscript more representative.